# Small Genomes, Big Disruptions: Parvoviruses and the DNA Damage Response

**DOI:** 10.3390/v17040494

**Published:** 2025-03-29

**Authors:** Rhiannon R. Abrahams, Kinjal Majumder

**Affiliations:** Institute for Molecular Virology and McArdle Laboratory for Cancer Research, University of Wisconsin, Madison, WI 53707, USA; rrabrahams@wisc.edu

**Keywords:** parvoviruses, DNA damage response, cell cycle regulation, gene therapy, oncolytic virotherapy

## Abstract

Parvoviruses are small, single-stranded DNA viruses that have evolved sophisticated mechanisms to hijack host cell machinery for replication and persistence. One critical aspect of this interaction involves the manipulation of the host’s DNA Damage Response (DDR) pathways. While the viral genome is comparatively simple, parvoviruses have developed strategies that cause significant DNA damage, activate DDR pathways, and disrupt the host cell cycle. This review will explore the impact of parvovirus infections on host genome stability, focusing on key viral species such as Adeno-Associated Virus (AAV), Minute Virus of Mice (MVM), and Human Bocavirus (HBoV), and their interactions with DDR proteins. Since parvoviruses are used as oncolytic agents and gene therapy vectors, a better understanding of cellular DDR pathways will aid in engineering potent anti-cancer agents and gene therapies for chronic diseases.

## 1. Introduction

Despite their small genome size and limited number of expressed viral proteins, parvoviruses have evolved sophisticated tools to usurp cellular transcription, replication, and DNA Damage Response (DDR) machinery for their benefit [1,2]. These interactions aid in the viral life cycle of autonomous parvoviruses like minute virus of mice (MVM) by facilitating the replication of the viral genome while simultaneously damaging the host in preparation for cell death, which releases progeny virions into the environment [3]. On the other hand, dependoviruses such as adeno-associated virus type 2 (AAV) cannot replicate on their own within the nuclear compartment; instead, they utilize cellular machinery to establish extrachromosomal viral genomes that form multimers ([4]; see Table 1 for comparisons). Our understanding of AAV biology is leveraged to engineer recombinant AAV (rAAV) gene therapy vectors. In the presence of co-infecting “helper” viruses such as adenovirus, herpesvirus, and even some related parvoviruses, AAV replicates in ways reminiscent of the autonomous parvoviruses, although the helper viruses have a significant impact on the cellular DDR factors [5]. This review will explore the recent discoveries on parvovirus infections and how they modulate the stability of the host genome. We will also compare the interactions between autonomous parvoviruses such as MVM and dependoviruses like AAV with cellular DDR pathways. Lastly, we will highlight some unresolved questions to which answers are needed to better understand the parvovirus life cycle, so that they can be leveraged to engineer safer gene therapy vectors and potent oncolytic agents.

## 2. Global Cellular DDR Pathways and Parvovirus Replication

The cellular DNA Damage Response (DDR) pathways are crucial for maintaining the fidelity of the genome in response to biotic and abiotic stress. Their absence can lead to the formation of DNA lesions that induce genome instability that can be toxic or carcinogenic. As obligatory intracellular parasites, viruses have evolved distinct means of usurping host DDR pathways for their benefit. These mechanisms vary widely, ranging from the induction of DDRs by polyomaviruses [6] and papillomavirus [7,8] which benefit the virus, to the inactivation of cellular DDR sensors by adenoviruses [9,10,11,12]. The parvovirus family, like other small DNA viruses like papillomaviruses and polyomaviruses, uses cellular DDR signals by leveraging its multifunctional non-structural proteins (see Table 1 for comparisons). The parvoviruses also deploy their single-stranded viral genomes for pathogenesis that are unique to *Parvoviridae*.

The DDR network comprises a tightly coordinated set of sensors, transducers, and effectors that work together to detect and repair DNA lesions on the host genome. Parvovirus replication is paradoxically correlated with the induction of cellular DDR signals mediated by an evolutionarily conserved family of Phosphatidyl-inositol-3-kinase-like-kinases [13]. While DDR signaling cascades phosphorylate a plethora of downstream proteins, their fundamental goal is to disrupt cellular functions such as transcription, replication, and cell cycle that afford the host cell enough time to repair DNA breaks and prevent mutagenic lesions from occurring. PI3-kinase-like kinases, including ATM (Ataxia-Telangiectasia Mutated), ATR (ATM and Rad3-related), and DNA-PK (DNA-dependent Protein Kinase), play a critical role in regulating the host DDR. Their activation is expected to hinder the viral life cycle by either inactivating viral genome transcription or integrating viral DNA into the host genome [13,14]. Strikingly, however, we and others have found that DDR signals are beneficial for *Parvoviridae* [15,16].

Extensive proteomic studies over the last three decades have characterized the cellular components that associate with the parvovirus non-structural proteins, using MVM, H-1PV, CPV, and AAV as model systems. These studies have deployed pulldown strategies during infection, ectopic expression, and marking proteins that are in close proximity using Biotin ligase prior to IP-Mass Spectrometry [17,18,19,20,21]. These studies have identified parvovirus non-structural protein binding partners belonging to cellular pathways involved in DNA replication, DDR signaling, gene expression, chromatin conformation, and cell cycle regulation. Many of these factors that interact with the viral non-structural proteins are directly involved in—or regulated by—ATM, ATR, or DNA-PK signaling. While mass-spectrometry-based studies have provided critical insights into the biochemical partners of parvovirus non-structural proteins, it remains unknown how these interactions create an environment that is favorable for the viral life cycle while also causing cellular toxicity. This parvovirus-induced DDR and subsequent cellular toxicity is the basis for its oncolytic capacity, extending to different cell types including primary cells, transformed cells, and even stem cells [22,23,24,25].

### 2.1. Parvovirus Replication Centers

The DDR network consists of a well-coordinated arrangement of sensors, transducers, and effectors collaborating to identify and repair DNA damage. It is likely that parvovirus genomes are also recognized by these proteins as potential lesions. Indeed, replicating parvovirus genomes such as AAV, MVM, H-1PV, and HBoV have been found to colocalize with cellular DDR proteins within the nuclear viral replication centers ([15,26,27,28,29,30,31]; see Table 1 for comparisons). This includes DDR sensors such as MRE11 and RPA; transducers such as ATM, ATR, and DNA-PK; and downstream effectors such as CHK2, CHK1, and P53 (Figure 1), leading these replication centers to be named Autonomous Parvovirus-Associated Replication (APAR) bodies [26,27,29,30]. However, it remains unknown whether these host DDR factors associate with viral genomes or were responding to virus-induced local DDRs, or both. Super-resolution imaging, biochemical fractionation, and genomics studies revealed that autonomous parvoviruses like MVM partially induce both phenomena that are described below [15,32].

### 2.2. Parvovirus Replication and ATM Signaling

Once DNA damage is detected by a sensor like MRE11, it activates a sophisticated multi-protein program to repair the incurred break. Both autonomous (MVM, HBoV, H-1PV) and dependo- (AAV2) parvovirus infection can activate several repair mechanisms that are typically involved in resolving DNA breaks and maintaining genome stability, including ATM and ATR kinases (Figure 1). ATM is primarily activated in response to DSBs (double-strand breaks), while ATR is activated in response to replication stress and single-stranded DNA (ssDNA; described in Section 2.3). Both ATM and ATR are serine/threonine kinases that phosphorylate numerous downstream targets, including key proteins involved in cell cycle regulation, DNA repair, and apoptosis (reviewed in [13]). DSBs are recognized by the MRN complex (MRE11-RAD50-NBS1), which recruits proteins that lead to the activation of ATM. Once activated, ATM phosphorylates critical targets like p53, BRCA1, and H2AX, coordinating the DDR and activating cell cycle checkpoints to prevent the progression of cells with damaged DNA (Figure 1). γH2AX, a phosphorylated form of histone H2AX, marks the site of DSBs to recruit cellular repair factors to restore the broken DNA molecule [33].

Treatment of MVM-infected cells with a chemical inhibitor to ATM kinase attenuates MVM replication, suggesting that the viral life cycle depends on ATM-mediated DDR signals [34]. Consistent with these observations, proteins in the ATM signaling pathway such as ATM, MRE11, RAD50, NBS1, BRCA1, and CHK2 have been found to colocalize with NS1 in APAR bodies [30,34]. The ability of ectopically expressed MVM-NS1 and AAV-Rep 68/78 to localize to cellular sites of DNA damage induced by laser micro-irradiation was not impacted by ATM inhibition [35,36]. These observations suggest that there are infection-specific outcomes of ATM signaling that are distinct from NS1′s association with cellular DDR sites that remain unknown.

The activation of ATM-mediated signals induces cellular DNA repair via two pathways: error-prone Non-Homologous End Joining (NHEJ) or error-free Homologous Recombination (HR) [37]. Since HR is active during the S and G2 phases of the cell cycle (when MVM replicates) due to the existence of a second copy of the cellular genome as a repair template, MVM likely interacts with HR pathway proteins. However, it is noteworthy that components of the NHEJ pathway, including DNA-PK and 53BP1, also associate with APAR bodies [34]. Using reporter cells that fluoresce upon HR or NHEJ-based repair outcomes, it has recently been shown that MVM infection inhibits the cellular HR pathway that facilitates efficient viral genome replication [38]. These findings are analogous to the programs deployed by high-risk HPVs to inhibit cellular HR [39]. In both cases, the viral non-structural proteins diminish the function of RAD51, impairing the cellular repair processes. Since RAD51 is associated with RPA-bound cellular single-stranded DNA [40], this might serve as the nexus between ATM and ATR signaling (described below).

### 2.3. Parvovirus Infection and ATR Signaling

As described earlier and reviewed extensively elsewhere, the ATR kinase pathway and CHK1 signaling have evolved to protect the host from the genomic instability and lesions that are generated by single-stranded DNA nicks and aberrations in replisome functions [41]. Stalled replication forks can expose single-stranded DNA regions, which are recognized and coated by Replication Protein A (RPA, [42]). ATR recognizes RPA-coated ssDNA and phosphorylates its downstream target, CHK1. This leads to cell cycle arrest at the G1/S or S/G2 checkpoints, affording the cell enough time to repair the accrued DNA damage [13]. ATR also activates the RAD17 and RAD9 complexes, which stabilize replication forks to facilitate efficient repair.

Despite its nicking function that is required for efficient MVM replication and its ability to induce replication stress on the cellular genome through its interaction with NS1 [43,44], NS1 surprisingly also directs the inactivation of the cellular ATR kinase (and its downstream CHK1 signaling) pathways [45]. This is initiated by NS1 protein localizing to cellular DDR sites in an ATR-dependent manner [35]. Soon thereafter, NS1 coordinates the inhibition of RAD51, a key regulator of cellular HR-dependent repair. To carry out this function, NS1 inhibits cellular Casein Kinase 2 (one of its interaction partners) from phosphorylating and activating RAD9, a component of the RAD9-RAD1-HUS1 (9-1-1) complex (Figure 1). As a result, RAD9 is unable to phosphorylate and activate TOPBP1, which is essential for ATR activation. This leads to suppression of the ATR-mediated DDR pathways during MVM infection, likely exacerbating host genome instability [38,45]. Taken together, these activities might indicate that NS1 possesses the ability to both initiate the damage that activates ATR/CHK1 signaling [46] and simultaneously inactivate the transducer components that are required for this pathway to properly function, priming the genome to undergo DNA breaks. Additionally, NS1-mediated inhibition of HR pathways might also explain the inability of MVM genomes to form extrachromosomal concatemers like AAV and their inability to integrate into the host genome (however briefly before cell death). Importantly, induction of cellular DNA breaks serves a proviral function for the MVM life cycle [15], likely by creating a nuclear environment where replication and repair proteins useful for the virus are available.

In response to replication stress and DNA lesions, ATR-mediated signals license stress tolerance pathways such as those regulated by Pol η, Pol κ, and Pol ζ [47]. Parvovirus infection can trigger TLS polymerases, allowing viral replication to continue despite the presence of cellular DNA damage. This has been observed for B19V [48] and HBoV [49]. Indeed RNAi-mediated silencing of the TLS polymerases reduced B19V and HBoV replication by 25–40 percent [48,49]. However, it is unclear whether these polymerases exclusively amplify the viral genome or also contribute to host genome instability. The former could lead to mutations in progeny viruses, driving viral evolution, whereas the latter would drive the establishment of new viral replication centers in the nucleus due to genome instability.

### 2.4. Parvovirus Replication and DNA-PK Signaling

Regulated by DNA-PK signaling and active throughout most stages of the cell cycle, the NHEJ pathway is typically the first line of defense in repairing DSBs [50]. However, this is an error-prone process that can have mutagenic consequences, since it directly ligates broken DNA ends without using a homologous template [51]. As described earlier, components of the NHEJ pathway, including DNA-PK and 53BP1, have been found to associate with APAR bodies [34]. However, treatment of cells with a DNA-PK inhibitor did not impact MVM replication [34] or alter the localization of NS1 to cellular DDR sites [35]. Interestingly, DNA-PK is activated by AAV replication in the presence of Adenovirus helper proteins [52]. This suggests that autonomous parvoviruses (MVM, HVoB, and H1-PV) and dependoparvoviruses (AAV) interact with DNA-PK-mediated signaling using distinct strategies (see Table 1 for comparisons).

## 3. Association of DDR Proteins with Parvovirus Genomes

### 3.1. Consequences of Parvovirus-Induced DDR and Immune Activation

The activation of DDR pathways by parvovirus infection is not without consequences. While DDR activation helps resolve DNA damage and allows viral replication, it can also lead to host-genome instability, cell cycle arrest, and cell death, all of which have implications for both the host and the virus. Incomplete or error-prone repair of DNA lesions can result in chromosomal rearrangements, deletions, and mutations [53]. Viral replication stress worsens these errors, leading to additional DNA breaks and new virus replication sites [15,54]. Parvoviruses like MVM have been shown to induce G2/M arrest [55], allowing time for the host genome to undergo repair or the viral genome to amplify in the nucleus [34]. However, if the damage is irreparable, the cell may undergo apoptosis to prevent the propagation of defective cells. These processes have been extensively characterized for the rodent autonomous parvovirus H-1PV, where NS1 expression and viral genome amplification cause cytopathic effects associated with cell death, DNA damage induction, chromosomal aberration, and generation of reactive oxygen species (ROS) [56]. Importantly, however, the balance between cell cycle arrest and cell death can influence the outcome of parvovirus infection, and the virus may manipulate these pathways to favor its replication, often by evading apoptotic mechanisms.

Parvovirus-induced DNA damage or viral genomes can also activate innate immune responses. DNA sensors such as Toll-Like Receptor 9 (TLR9) contribute to the activation of type-I interferon signaling that inhibits MVM and H-1PV parvoviruses. This might be a tissue-specific outcome, because TLR9s in plasmacytoid dendritic cells do not detect MVM [57]. Interestingly, transformed cells that lack these DNA sensors or signaling pathways are prime targets for MVM/H-1PV-mediated cell death [58]. Alternately, MVM/H-1PV may have evolved distinct strategies to evade the inhibitory action of IFNs on the host environment [59]. These properties enhance the exciting potential for using rodent protoparvoviruses as direct oncolytic agents or as enhancers of cancer immunotherapies.

### 3.2. Regulation of Parvovirus Gene Expression by DDR Factors

Autonomous parvovirus replication requires S-phase entry by the host cell, presumably to use the cellular DNA polymerase delta to amplify the viral genome [60]. These viral genomes require viral proteins for genome processing, synergizing with host factors to form the nuclear condensates referred to as APAR bodies. APAR bodies have been monitored by staining for the MVM non-structural protein NS1, which possesses trans-activation, helicase, and ATP-ase activities [61]. By virtue of its ability to bind to ACCAACCA consensus motifs on the MVM genome, NS1 transcriptionally activates the viral P4 and P38 promoters [62,63], initiating a feed-forward loop that generates more and more NS1 proteins that transcriptionally regulate the progeny viral genomes. We have recently discovered that NS1 bound to the MVM genome transports the viral genome to cellular sites of DNA damage, presumably to jump-start virus replication [64]. As virus replication continues, the cell cycle status of the host further regulates the assembly of progeny virions [65].

As viral replication continues, the single-stranded form is converted into double-stranded, and subsequently into a dimer molecule [66]. MVM NS1 is essential for cleaving these dimer molecules into two monomers for replication to continue, remaining covalently attached to the 3′ end of the replicating MVM genome. This NS1-induced nick to the MVM genome might be the initiation point for recognition of the viral genome by cellular DDR sensors. Instead, we have recently discovered that the MVM genome is bound by MRE11 on the viral promoters P4 and P38 [67]. Interestingly, the other components of the MRN complex, RAD50 and NBS1, do not associate with the MVM genome, making it unlikely that MRN processes MVM [67]. Moreover, in the absence of MRE11, MVM replication is severely decreased, making it distinct from the repressive role played by MRE11 on the AAV genome [31,68,69,70]. These observations suggest that autonomous and dependoparvoviruses have evolved distinct mechanisms to usurp cellular DDR factors (see Table 1 for comparisons).

### 3.3. Transcriptional Regulation of Parvovirus-Induced Cell Cycle Defects

One of the hallmarks of parvovirus infection is the induction of cell cycle defects. This includes delays or arrests in specific cell cycle phases, particularly at checkpoints where DNA damage is assessed [71]. Parvoviruses like MVM, B19V, and CPV interfere with normal cell cycle progression by triggering DDR pathways that cause cell cycle arrest at the G2/M border [55,72,73]. These defects are not only important for viral replication—since phases of the cell cycle like S and G2 provide optimal conditions for viral genome replication—but also contribute to the pathogenesis of the virus. MVM replication is tightly linked to cell cycle progression, particularly in the S-phase, where cellular DNA replication is also ongoing.

Cyclin B is a key regulator of the G2/M transition in the cell cycle, and its activation is tightly controlled by cellular transcription factors that bind cell cycle genes to orchestrate proper cell division (reviewed in [74]). Parvoviruses such as MVM transcriptionally inhibit the expression of mitotic entry genes that are a part of the cellular DREAM complex [75]. Some of these factors, such as FOXM1, are precluded from binding to the cellular Cyclin B1 promoter (Figure 2), leading to a potent pre-mitotic cell cycle block at the G2/M border [76]. However, this interaction is not without consequences. The abnormal activation of cyclin B in the presence of DNA damage can result in incomplete DNA repair, leading to chromosomal instability and cell death [77]. These properties are beneficial for engineering potent oncolytic viruses. While CPV infection also induces a G2/M arrest, HBoV induces an S-phase cell cycle arrest through unknown mechanisms (Figure 2).

### 3.4. Induction of Host Cell Genome Instability by RPA Exhaustion

As a core constituent of APAR bodies, the single-stranded DNA binding protein RPA is tasked with recognising stalled replication forks, activating local ATR signaling cascades that lead to recruitment and activation of ATR, ATRIP, TopBP1, 9-1-1 complex, and eventually CHK1 [13,34,46]. We have recently discovered that the MVM genome serves as a sink for RPA molecules, depleting the host genome of this essential factor that is required to maintain genome stability [54]. Furthermore, during initial MVM replication, RPA binding leads to phosphorylation, recruitment, and activation of ATR kinase, which is critical for efficient MVM replication [35]. As infection progresses, MVM inactivates the ATR-CHK1 signaling axis [46]. This depletion of RPA from the host followed by inactivation of CHK1 signaling serves as a one-two punch, exacerbating host genome fragility to induce widespread cellular DNA damage. Like MVM, HBoV and CPV manipulate DDR pathways to regulate the cell cycle, ensuring an optimal environment for viral replication. HBoV depends on the ATR pathway, while CPV engages the ATM pathway more robustly. MVM, on the other hand, utilizes DDR to both promote replication and actively induce cell death, with strong ATM activation [34], eventual ATR inactivation [46], and Rad51 recruitment [45]. Using MVM as a model system, we have identified multiple transcriptional and replicative strategies deployed by parvoviruses to cause cell death. These mechanisms might also be generalizable to HBoV or CPV-induced apoptotic signals which also exploit p53-mediated signals to manipulate the host cell cycle and cause cell death (Figure 2). On the other hand, MVM triggers apoptosis via p53 activation to facilitate virion release. Distinct from MVM replication, the genomes of the dependoparvoviruses like AAV do not replicate on their own within the host-cell nucleus, remaining instead as an extrachromosomal concatemer long-term [78,79,80,81].

## 4. AAV Life Cycle and DDR Signals

### 4.1. Introduction to AAV’s Interaction with the Host DDR

Like other parvoviruses, AAV interacts with the host’s DNA damage response (DDR) machinery to ensure efficient replication [52], integration [82], and persistence [81] of its genome within the host nuclear compartment. AAV replication requires the Rep68/78 proteins, which bind to specific regions of the viral genome to regulate transcriptional activation and genome processing [83]. The Rep68/78 proteins possess helicase and endonuclease activities that create ssDNA intermediates and induce DNA breaks [21,52]. Since ssDNA intermediates and breaks activate ATR signaling, it is likely that ATR signaling regulates AAV replication. Consistent with this hypothesis, ATR inhibition partially attenuated AAV replication [84]. On the other hand, DNA-PK is enriched in cells lacking the ring finger protein RNF121, which regulates the transcriptional activation of AAV gene expression [85]. These observations suggest that DDR signals activate yet-to-be-discovered pathways that eventually regulate AAV gene expression.

In rare cases (approximately 0.1% of viral genomes), AAV can integrate into the host genome at chromosome 19q13.3 [86,87]. The Rep proteins are essential for site-specific integration, binding to both the viral genome and the host chromosomal DNA, which leads to double-strand breaks (DSBs) at the integration site [88,89,90]. The non-homologous end joining (NHEJ) pathway is critical for repairing these breaks and integrating the AAV genome into the host genome [91]. The formation of DSBs during AAV integration triggers local ATM signals, subsequently activating the NHEJ repair pathway. NHEJ proteins such as KU70/80 and DNA-PKcs then repair these breaks and facilitate the stable integration of the AAV genome into the host chromosome [91]. However, it is noteworthy that lack of ATM enhances rAAV transduction and integration, likely invoking other modes of DNA repair [92]. In this regard, homologous recombination (HR) can also play a role, especially during the use of gene therapy vectors that integrate into dividing cells [82]. These observations are surprising because no DNA sequences homologous to AAV have been identified in the mammalian genome. An alternative possibility might be the integration events following Rep 68/78-mediated DNA break induction [93,94,95]. Indeed, the AAV integration sites are known to be in proximity to Rep-binding sequences [96,97]. Additionally, PARP1, a protein that repairs host single-strand breaks, is also involved in AAV genome integration [98]. Its role in stabilizing DNA replication forks and interacting with DNA ligase indicates it might regulate the DDR machinery needed to integrate AAV genomes [99].

The ability of AAV to persist as an episome or integrate into the host genome depends on its engagement with DDR pathways. While ATM and NHEJ repair mechanisms facilitate site-specific integration, the ATR signaling pathway may be more critical in managing AAV concatemer formation and genome stability (Figure 3). The persistence of the AAV is vital for its use in gene therapy. Unlike other viral vectors such as lentiviruses, which integrate randomly into the host genome, AAV preferentially integrates into the AAVS1 site on chromosome 19 or persists as an episome, minimizing genomic instability. This controlled interaction with DDR pathways, particularly through ATM-mediated NHEJ repair, allows AAV to establish long-term gene expression without the risk of insertional mutagenesis. Recombinant AAV (rAAV) vectors used in gene therapy are further tailored to encourage episomal persistence over integration, enhancing their safety profile for clinical applications.

### 4.2. Manipulation of DDR Signals to Benefit AAV

AAV genomes enter the nucleus in the form of a single-stranded DNA molecule, which is then processed into a partially double-stranded intermediate that resembles a stalled replication fork [100], ultimately recombining with other AAV genomes to form a multimeric episome that persists long-term [78,79,101]. This state is sustained by the DDR and host chromatin machinery, preserving the episome long-term. Indeed, pioneering studies using dual vector systems have recently identified members of the HR pathway, including RAD51 and BRCA1 (Figure 1), in regulating the formation of AAV concatemers [81]. In embryonic stem cells with a distinct chromatin landscape, these AAV genomes can activate global p53 signals, causing genome instability that leads to cell death [102]. While the AAV-ITR contains p53 binding sequences, the specific triggers for p53 activation in ES cells remain unclear. MRE11 represses AAV gene expression and rAAV transduction, possibly by recruiting repressive chromatin modifiers to the viral genome, the identity of which remains unknown [68,69]. Importantly, DDR protein-mediated recruitment of host chromatin modifiers to transcriptionally repress local gene expression [103] might also function in regulating AAV gene expression. Indeed, chromatin modifiers like the HUSH complex have a transcriptionally repressive effect on AAV gene expression [104]. Analogous to these observations, MRE11 has been found to recruit the histone methyltransferase LSD1 to cellular telomeres, but it remains unknown how MRE11 might regulate viral gene expression [105].

Perhaps this is why successful AAV replication necessitates a co-infecting helper virus, such as adenovirus or herpesvirus, which inactivate MRE11 and its downstream repressive signals on the AAV genome [9,69,106]. Based on their interaction with MRE11, we propose that the parvovirus family has evolved distinct mechanisms of dependence (autonomous parvoviruses) and independence (dependoparvoviruses). Still, the presence of various helper viruses, including adenovirus (Ad), herpes simplex virus (HSV), or radiation-induced stress can significantly alter how AAV interacts with these DDR pathways (described below). Each of these stressors induces unique forms of DNA damage, leading to differential activation of DDR signaling networks that, in turn, influences AAV’s replication and genome maintenance. How helperviruses aid AAV-induced DDR signals is summarized below.

#### 4.2.1. Adenovirus-Mediated Manipulation of DDR for AAV Replication

Adenovirus (Ad) was the originally discovered helper virus that aided the AAV life cycle [107], providing the factors essential for AAV replication, transcription, and packaging [108]. Indeed, Ad capsid proteins have been suggested to aid the translocation of AAV particles into the host-cell nucleus [109]. Upon nuclear entry, Ad replication induces cellular double-strand breaks (DSBs) that are associated with remodeling of the host nuclear proteome [110,111,112]. Adenovirus-induced DSBs inactivate the ATM pathway by degrading MRE11 [9,10,113] and also inactivate ATR signaling (Figure 3, [114]). Surprisingly, during replicative AAV infection, Adenovirus helper proteins stimulate ATM, ATR and DNA-PK-mediated cellular signals [52,84]. However, silencing of ATM and DNA-PK signaling using chemical inhibitors and RNAi in the presence of the Ad helper proteins decreased AAV replication by 50%, but ATR inhibition attenuated AAV replication by almost 90% [84]. Since Rep68/78 are known to associate with cellular DDR sites [36], it is conceivable that these interactions also facilitate efficient AAV replication in conjunction with cellular DDR induction (described below for radiation-induced AAV replication).

The adenovirus helper factors that are essential for AAV replication are E1A, E1B55K, E2A, E4orf6, and VA-RNA [5]. The helper function of E1A is to activate the AAV P5 promoter by recruiting chromatin modifiers and transcriptionally drive cells into S-phase [115]. E2A-DBP, being a single-stranded DNA binding protein, likely regulates AAV genome processing and splicing of Rep transcripts [116]. E4orf6 forms a complex with E1B55K to promote second-strand synthesis of AAV, likely by degrading and mislocalizing MRE11 to inhibit MRN activation [9,114,117]. Lastly, VA RNA prevents E4orf6/E1B-mediated degradation of the AAV capsid and Rep52 [118]. These factors enhance viral transcription and likely aid AAV in evading the host immune system during replication.

#### 4.2.2. HSV-Mediated Manipulation of DDR for AAV Replication

Herpes simplex virus (HSV) deploys a pathogenic program that is distinct from that of the adenovirus life cycle in the nucleus of a host cell but can still serve as a helper virus that facilitates AAV replication [119]. However, the interaction between AAV and DDR is more intricate in the presence of HSV due to its characteristic latent and lytic stages of viral infection [120]. The HSV lytic cycle triggers double-strand breaks (DSBs) and replication stress, primarily caused by the large viral genome and the formation of viral replication intermediates [121,122,123,124]. The replication of HSV DNA inactivates ATR/Chk1 signaling, which is like the ATR inactivation observed upon Adenovirus infection (Figure 3, [114,121]).

HSV provides transcription factors and helper proteins, such as ICP4 and ICP27, which enhance AAV transcription [125]. However, the replication stress caused by HSV can create challenges in AAV genome replication. Unlike Ad, which directly fosters AAV genome replication, HSV-induced replication stress may lead to reduced replication efficiency or delays in viral genome replication. Furthermore, unlike adenovirus infection, which inactivates MRE11, HSV-mediated AAV replication is enhanced by host MRE11 [70]. Perhaps related to this, HSV infection facilitates AAV genome integration into AAVS1 through Rep binding sites. Since the HSV genome also contains Rep binding sequences, it remains unclear whether AAV integrates into HSV and if MRE11 controls this process [126]. The episomal persistence of AAV during latent HSV infection is also influenced by the virus’s ability to modulate host DNA repair mechanisms and immune surveillance, much of which remains unknown.

#### 4.2.3. Radiation-Induced Manipulation of DDR for AAV Replication

Radiation induces double-strand breaks (DSBs) and single-strand breaks (SSBs), activating all PI3-kinase-like kinases and fork stalling. Accumulation of excessive DNA damage can exceed the cell’s repair capacity, resulting in genomic instability. After radiation exposure, AAV can leverage the induced DSBs for genome replication even in the absence of helper virus-induced DNA damage [16]. However, the NHEJ repair pathway plays a more significant role in AAV replication during radiation-induced damage. HR (homologous recombination) is less active in non-dividing cells, which makes NHEJ a crucial regulator in responding to AAV-associated DNA damage. Unlike Ad or HSV, which provide viral factors to enhance AAV replication and integration, radiation-induced DSBs are solely host-mediated responses, possibly accounting for the inefficient replication of AAV in these conditions. It is noteworthy that AAV replication in all cases (presence or absence of helper virus) is ultimately associated with cell death. Supporting this observation, autonomous parvoviruses such as HBoV are sufficient to facilitate AAV replication, possibly by inducing cellular DNA damage, but not as efficiently as AdV or HSV.

## 5. Association of Parvovirus Genomes with Cellular DDR Sites

Despite extensive characterization of the cellular DNA repair, replication, and transcriptional proteins that are associated with APAR bodies, it is still unknown where in the nuclear compartment these condensates are formed. Using a modified iteration of chromosome conformation capture assay coupled with high-throughput sequencing, we have discovered that MVM genomes associate with pre-existing cellular sites of DNA damage [15,127]. Initial phases of MVM replication exacerbate the DNA damage at these genomic regions before causing wide-spread chromosomal DNA damage. Interestingly, these cellular DDR sites correlated with fragile genomic regions that are generated by replication–transcription conflicts [128,129,130,131] and are sites where other small DNA viruses like human papillomavirus also localize [132,133]. Surprisingly, these cellular DDR sites and MVM-associated regions are packaged into transcriptionally accessible Type-A chromatin that forms distinct subnuclear topological structures [15,134]. Building on these findings, using ChIP-seq assays, we have discovered that the NS1 protein bound to the MVM genome facilitates the recruitment of the viral genome to these cellular DDR sites [64]. The NS1 protein is additionally capable of recruiting heterologous DNA molecules containing the ACCAACCA consensus motifs to induce cellular DDR sites [64]. These findings that NS1 aids in DNA transport add to the growing list of its functions in the MVM life cycle. NS1 appears to additionally inhibit ATR/CHK1 signaling by redirecting key transducer components like CK2 [45], ultimately priming the genome for DNA breaks. However, despite NS1′s ability to transport MVM to DDR sites, the viral genome has not been found to integrate into the host, instead replicating vigorously as extrachromosomal monomer-dimer intermediates in proximity to the cellular DDR sites [135]. This inability of MVM to integrate into the host might point to another function of NS1: inhibiting homologous recombination pathways (described above).

Since ectopically expressed Rep 68/78 proteins of AAV have been shown to induce S-phase arrest in host cells by activating CHK1-dependent cellular signals while interacting with cellular DDR sites [136], we predicted that AAV genomes would also localize with cellular DDR sites in the same way as MVM. However, while AAV genomes were associated with the vicinity of cellular DDR sites, these regions were not associated with broad chromosomal domains or territories that were observed with MVM [36]. Recombinant AAV (rAAV) genomes did not localize to these cellular DDR sites, suggesting the AAV-ITR elements are not sufficient to direct the localization of the viral genome [36]. Instead, the rAAV genomes accumulate in subnuclear compartments called nucleoli [137], which are associated with expression of ribosomal genes. However, it remains unknown whether the nucleoli-associated rAAV genomes are the vector genomes that are the expression platforms for rAAV transgenes.

## 6. Additional Mechanisms of Cellular DDR Activation by Parvoviruses

Characterization of the cellular replication forks generated in host cells during MVM replication revealed an increase in new origin firings. This might be caused by the inactivation of the ATR signaling pathway by NS1 (described above). Importantly, inhibition of replication origin firings leads to a decrease in virus replication and virus-induced cellular DDR [54]. MVM seems to have evolved redundant mechanisms to perturb cellular replication forks, including the degradation of P21 through CRL4Cdt2-mediated pathways (which subsequently impacts PCNA stability), destabilizing primase from polymerase alpha [138,139] to accentuate replicative stress, and via activation of mitochondrial ROS (which has been characterized for the related H-1PV rodent parvovirus [56]). Interestingly, mitochondrial ROS signals induce ATM activation, which might have a pro-viral effect on the MVM life cycle [140].

## 7. Outstanding Questions

Activation of innate immune signals in the host and how they might be activated—perhaps DNA molecules from the replisome or broken cellular DNA feed back into the cytoplasm to activate local immune signals such as cGAS/STING [141]. Some of these signals, like that of STAT signaling, are utilized by parvoviruses such as B19V via replication factors like MCM helicase [142]. MVM-mediated interactions with the cellular immune signaling pathways may similarly be leveraged to engineer potent oncolytic agents [143]. However, the extent to which dependovirus-induced DNA breaks and replication stress affect long-term genome stability remains unknown. It is critical to understand how dependoviruses interact with the host in ways that are distinct from those of autonomous parvoviruses (MVM, MVC, CPV) that generate extensive DNA damage [34,144,145]. A better understanding of these differences will aid in the engineering of better oncolytic virotherapies as well as safer gene therapy vectors. Beyond therapeutic interventions, these studies will also shed light on improving gene therapy vector production outcomes for widespread clinical usage.

Parvoviruses exemplify how minimal viral genomes can commandeer host cellular processes through sophisticated manipulation of DNA damage response pathways. By activating and subverting key DDR components such as ATM, ATR, and associated repair factors, these viruses not only ensure their replication but also profoundly influence host cell fate, genome stability, and immune activation. This intricate interplay between viral factors and host repair machinery opens exciting avenues for therapeutic applications, including oncolytic virotherapy and gene therapy, while also raising critical questions about the long-term genomic consequences of viral genome persistence (caused by rAAV vectors). Future research into the molecular interactions and regulatory networks at the virus–host interface will be essential for understanding viral pathogenesis and developing clinical applications.

## Figures and Tables

**Figure 1 viruses-17-00494-f001:**
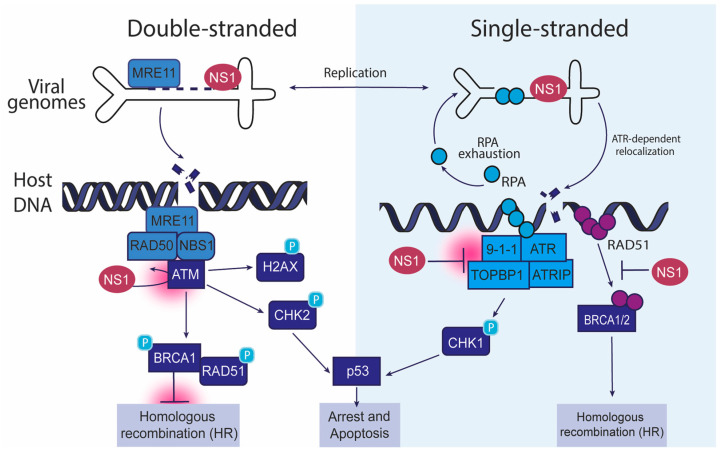
Interaction of autonomous parvoviruses with cellular DDR factors. This schematic illustrates how autonomous parvoviruses, such as MVM, HBoV, and CPV, manipulate host DDR pathways during infection. The left panel depicts the interaction of double-stranded viral genomes with the ATM (ataxia telangiectasia mutated) pathway. The right panel shows single-stranded viral genomes engaging the ATR (ataxia telangiectasia and Rad3-related) pathway. In the context of double-stranded DNA, the viral NS1 protein localizes to DNA damage, leading to the recruitment of the MRE11-RAD50-NBS1 (MRN) complex, and activates ATM kinase. This causes the phosphorylation of downstream effectors such as H2AX, CHK2, and BRCA1, promoting homologous recombination (HR). Sustained ATM activation can cause p53-dependent cell cycle arrest and apoptosis. In the right panels, MVM-induced replication stress is caused by RPA exhaustion, which triggers the ATR pathway through the 9-1-1 complex, TOPBP1, and ATRIP, resulting in CHK1 phosphorylation. ATR signaling and RAD51 recruitment leading to homologous recombination are collectively inhibited by NS1. These interactions highlight how autonomous parvoviruses exploit double-stranded and single-stranded DDR pathways to enhance their replication, modulate host genome stability, and influence cell fate decisions.

**Figure 2 viruses-17-00494-f002:**
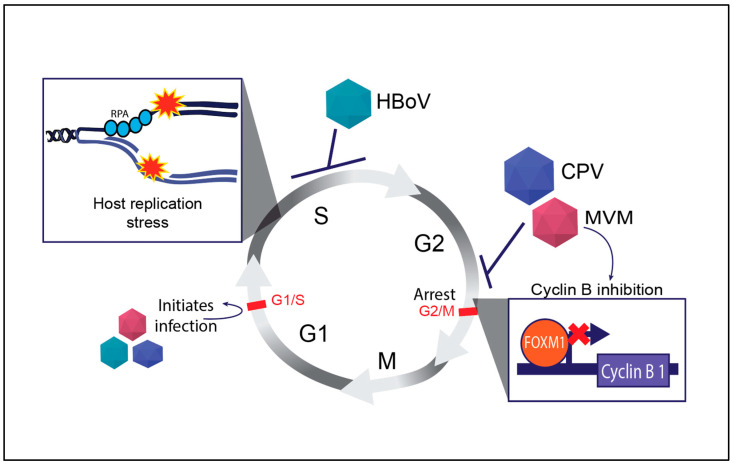
Regulation of mammalian cell cycle by autonomous parvoviruses. The key cell cycle checkpoints are indicated by red text and bars. The inhibition of the cell cycle by distinct parvoviruses is indicated by blunt arrows and virally activated processes are indicated by normal arrows. The inset shows the molecular outcomes of parvovirus infections on host replication forks (top left) and expression of cell cycle regulator genes like Cyclin B1 (bottom right). Abbreviations used: Replication Protein A (RPA), Human Bocavirus 1 (HBoV), Canine Parvovirus (CPV), Minute Virus of Mice (MVM).

**Figure 3 viruses-17-00494-f003:**
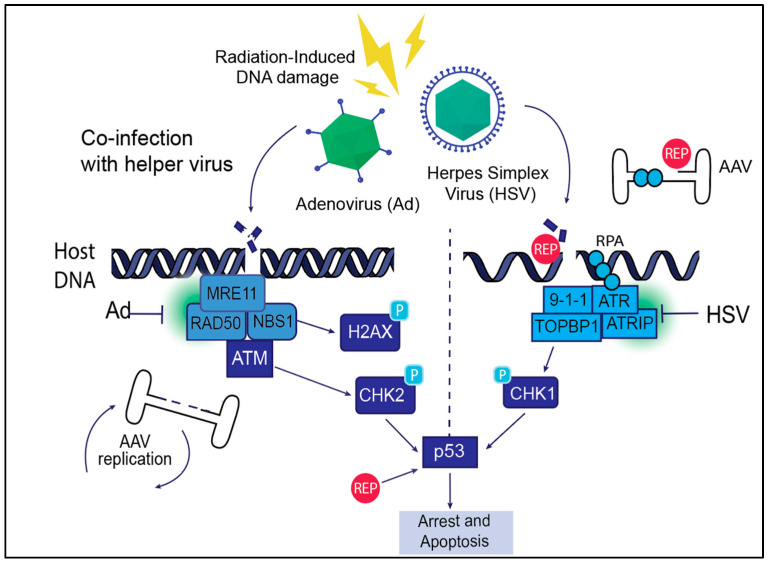
AAV life cycle and DNA Damage Response (DDR) signaling. AAV exploits host DDR machinery during its life cycle to facilitate efficient replication, genome integration, and persistence within host cells. The AAV Rep proteins Rep68/78 induce DNA breaks and create ssDNA intermediates that activate the ATR signaling pathway. Helper viruses such as adenovirus (Ad) and herpes simplex virus (HSV) manipulate host DDR signaling to enhance AAV replication and integration. Helper viruses induce both double-stranded (**left panel**) and single-strand (**right panel**) breaks on the host. Ad promotes AAV replication through interaction with the ATR pathway, and HSV causes replication stress and interaction with ATR that can delay replication but also support genome integration. Radiation-induced DNA damage similarly activates DDR signaling, facilitating AAV genome replication and integration, though AAV must rely more on host DDR machinery in this context. Together, these interactions highlight the critical role of DDR pathways in supporting AAV replication and persistence.

**Table 1 viruses-17-00494-t001:** Comparative overview of key parvovirus interactions with DDR. This table summarizes the similarities and differences among four parvoviruses—minute virus of mice (MVM), human bocavirus 1 (HBoV1), canine parvovirus (CPV), and adeno-associated virus (AAV). It highlights their replication strategies (autonomous vs. helper-dependent), cell cycle targeting (S-phase cells), manipulation of DNA Damage Response (DDR) pathways, and the roles of NS1 or Rep proteins in inducing DNA breaks and modulating host repair factors. The table also covers their potential in oncolytic virotherapy or gene therapy, viral persistence strategies, impact on host genomic integrity, and disease associations, illustrating how these features converge and diverge across the parvovirus family.

Category	MVM	HBoV1	CPV	AAV
Helper virus dependence	Autonomous, does not require a helper virus			Requires a helper virus for efficient replication
Manipulation of DDR	Activates ATR/ATM, p53, CHK1 for replication			Activates ATR/ATM, p53, CHK1 but relies on a helper virus
NS1/Rep68/78 Role in DDR	NS1 induces DNA breaks, interacts with Rad51, Chk1			Rep68/78 induces DNA breaks, interacts with Rad51, Chk1
Oncolytic/Gene Therapy Potential	Potential due to DDR manipulation			Widely used in gene therapy due to stable genome persistence
Cell Fate Decisions	Leads to senescence in many cases	Can establish persistent infection or chronic inflammation		Can persist without causing immediate cell death
Viral Persistence	Less likely to persist, often leads to senescence or cell death	More likely to establish latent/persistent infection		Can integrate into AAVS1 site on chromosome 19 or as an extrachromosomal episomes
Impact on Genomic Integrity	Causes genome instability by inhibiting HR	Generates double-strand breaks repaired via HR/NHEJ		Can integrate at a specific site, minimal genomic instability
Disease Association	Infects mice, used as a model virus	Infects humans, causes respiratory tract infections	Infects dogs, causes severe gastrointestinal disease	Non-pathogenic in humans, used for gene therapy

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
