# Peer review of "Small Genomes, Big Disruptions: Parvoviruses and the DNA Damage Response"

_viruses, 2025, doi:10.3390/v17040494_

Round 1
Reviewer 1 Report
Comments and Suggestions for Authors
Review
The Review article entitled “Small genomes, big disruptions: parvoviruses and the DNA damage response” contributed by Abrahams and Majumder, provides a comprehensive and updated review on the complex interactions with the host DDR of three representative members of the Parvoviridae. The review is consistent with the important efforts on this issue that the K. Majumder´s lab is carrying out over the last few years.
The article in the presented version is well structured and supported by very useful and well-illustrated figures.
The review is in extensive format and in considerable depth. This gives a well-founded insight into the complexity and diversity of Parvovirus interactions with the cellular DDR machinery. But it also hints at the need for novel multiple experiments in different virus-host cell systems to fully understand the mechanisms involved and to exploit them towards the therapeutic applications of these viruses.
A few Points for consideration of the authors to potentially improve their message:
- Please complete the missing References 3 and 6
- There are other cell cycle-regulated processes of the MVM infection that may affect its pathogenesis, such as the nuclear transport of capsid proteins in the S-phase. This could be mentioned in the section addressed between lines 234-241 of the manuscript.
- It may also be interesting to mention with specific reference that the DDR is induced by some parvoviruses not only in established cell lines in culture but also in transformed human primary and even stem cells, which may be an essential property of parvovirus oncolytic capacity.
Author Response
Comment 1: Please complete the missing References 3 and 6.
Response: We have corrected these references (line 576 and 581).
Comment 2: There are other cell cycle-regulated processes of the MVM infection that may affect its pathogenesis, such as the nuclear transport of capsid proteins in the S-phase. This could be mentioned in the section addressed between lines 234-241 of the manuscript.
Response: Thank you for the suggestion. We have made the relevant change in the manuscript (line 239-241).
Comment 3: It may also be interesting to mention with specific reference that the DDR is induced by some parvoviruses not only in established cell lines in culture but also in transformed human primary and even stem cells, which may be an essential property of parvovirus oncolytic capacity.
Response: Thank you for the suggestion. We have made the addition with relevant references to this section (line 89-91).
Reviewer 2 Report
Comments and Suggestions for Authors
In this review manuscript Abrahams and Majumder summarize the mechanisms through which parvoviruses of various replication strategies utilize and exploit the DNA damage response of the host cell. The review walks the reader through the pathways that are involved in addressing intranuclear DNA damage and then describes the specifics of what the virus-host interplay looks like at viral genome-viral and cellular proteins, viral proteins - cellular proteins and cellular DNA -viral proteins levels. It also devotes a section to these mechanisms in the presence of a helper virus, as seen for adeno-associated viruses. Overall, the manuscript is well-written, provides a detailed, but not overly dense guide to the field, with plenty of relevant citations. Only a couple of minor issues persist, which the authors should address:
-Throughout section 2.1, 2.2, 2.3: I realize that this was written as a general review of the topic, but just simply saying "parvoviruses" throughout is not specific enough. The exact virus should always be specified for which these findings were established, as "parvoviruses" is a very broad term, encompassing all members of an extremely diverse family and there is no evidence that these mechanisms are conserved even in e.g. densoviruses.
- Virus names throughout should be written in lower case initials, such as adeno-associated virus, parvovirus, adenovirus, etc.
- Line 193: NS1 expression is also part of viral infection. Which steps of viral infection are being referred to here?
-Line 217: "more viral genomes" - does this refer to the new progeny genomes?
- Line 257: "is asked" - I assume this is "is tasked"?
- Figure 2: as the figure should be possible to comprehend independently from the main text, the abbreviations should be resolved in the caption. Which HBoV is referenced here?
- Line 277: Which nuclear compartment?
- Table 1: there are four different HBoV types, with rather distinct pathology. It should be specified which one is discussed here. Same for AAV, both here and throughout. Recently, pathology has been associated with AAVs, which should be mentioned here as well.
- Line 304: The exact frequency of the integration should be specifically stated here. "In rare cases" is too vague.
- Line 367: The AdV helper also contributes to AAV virion translocation to the nucleus
- Line 380: E1 and E2 are not proteins, they are transcription units of the AdV genome, which express several proteins. This should be corrected.
- Line 382: same issue as above; the protein referenced here is the DBP of the E2A early expression unit
Author Response
Comment 1: Throughout section 2.1, 2.2, 2.3: I realize that this was written as a general review of the topic, but just simply saying "parvoviruses" throughout is not specific enough. The exact virus should always be specified for which these findings were established, as "parvoviruses" is a very broad term, encompassing all members of an extremely diverse family and there is no evidence that these mechanisms are conserved even in e.g. densoviruses.
Response: Thank you for the suggestion. We have gone through the section and specified the type of parvovirus being discussed in the relevant text (line 96, 108, 194).
Comment 2: Virus names throughout should be written in lower case initials, such as adeno-associated virus, parvovirus, adenovirus, etc.
Response: Thank you for the suggestion. We have corrected the virus names from upper-case to lower case throughout the manuscript (line 23, 26, 29, 33, 306, 307, 308, 405, 450, 451, 465).
Comment 3: Line 193: NS1 expression is also part of viral infection. Which steps of viral infection are being referred to here?
Response: We are referring to NS1 expression and viral genome amplification, which we have clarified in the text (line 210).
Comment 4: Line 217: "more viral genomes" - does this refer to the new progeny genomes?
Response: Yes. We have specified progeny viral genomes (line 237).
Comment 5: Line 257: "is asked" - I assume this is "is tasked"?
Response: Yes. This has been corrected (line 282).
Comment 6: Figure 2: as the figure should be possible to comprehend independently from the main text, the abbreviations should be resolved in the caption. Which HBoV is referenced here?
Response: Thank you for the suggestion. We have provided the abbreviations in the caption. The HBoV references here is HBoV1, which we also specify in the caption (Fig. 2).
Comment 7: Line 277: Which nuclear compartment?
Response: The host cell nucleus, which we now specify (line 302).
Comment 8: Table 1: there are four different HBoV types, with rather distinct pathology. It should be specified which one is discussed here. Same for AAV, both here and throughout. Recently, pathology has been associated with AAVs, which should be mentioned here as well.
Response: All of the HBoV described here is HBoV1, which we now specify in the table. For the purpose of simplicity, we have restricted our discussions about AAV-DDR interactions also to AAV2. We have described this in the manuscript (table 1).
Comment 9: Line 304: The exact frequency of the integration should be specifically stated here. "In rare cases" is too vague.
Response: We have specified the frequency and provided a relevant primary citation (line 348).
Comment 10: Line 367: The AdV helper also contributes to AAV virion translocation to the nucleus
Response: Thank you. We have made this addition with the relevant citation (line 414,415).
Comment 11: Line 380: E1 and E2 are not proteins, they are transcription units of the AdV genome, which express several proteins. This should be corrected.
Response: Thank you for the suggestion. We have corrected this error (line 426, 427).
Comment 12: Line 382: same issue as above; the protein referenced here is the DBP of the E2A early expression unit
Response: We have also corrected this error (line 429).